# Coexistence of structural and magnetic phases in van der Waals magnet CrI₃

Jaume Meseguer-Sánchez[1], Catalin Popescu [2], José Luis García-Muñoz [3], Hubertus Luetkens [4], Grigol Taniashvili[5], Efrén Navarro-Moratalla [1✉], Zurab Guguchia [4✉] & Elton J. G. Santos [6,7✉]

CrI₃ has raised as an important system to the emergent field of two-dimensional van der Waals magnetic materials. However, it is still unclear why CrI₃ which has a ferromagnetic rhombohedral structure in bulk, changed to anti-ferromagnetic monoclinic at thin layers. Here we show that this behaviour is due to the coexistence of both monoclinic and rhombohedral crystal phases followed by three magnetic transitions at $T_{C1} = 61$ K, $T_{C2} = 50$ K and $T_{C3} = 25$ K. Each transition corresponds to a certain fraction of the magnetically ordered volume as well as monoclinic and rhombohedral proportion. The different phases are continuously accessed as a function of the temperature over a broad range of magnitudes. Our findings suggest that the challenge of understanding the magnetic properties of thin layers CrI₃ is in general a coexisting structural-phase problem mediated by the volume-wise competition between magnetic phases already present in bulk.

[1] Instituto de Ciencia Molecular, Universitat de Valéncia, Paterna, Spain. [2] CELLS-ALBA Synchrotron Light Facility, Cerdanyola del Valles, Barcelona 08290, Spain. [3] Institut de Ciència de Materials de Barcelona (ICMAB), CSIC, Bellaterra, Catalunya, Spain. [4] Laboratory for Muon Spin Spectroscopy, Paul Scherrer Institute, Villigen PSI, Switzerland. [5] Department of Physics, Tbilisi State University, Tbilisi, Georgia. [6] Institute for Condensed Matter Physics and Complex Systems, School of Physics and Astronomy, The University of Edinburgh, Edinburgh EH9 3FD, UK. [7] Higgs Centre for Theoretical Physics, The University of Edinburgh, Edinburgh EH9 3FD, UK. ✉email: efren.navarro@uv.es; zurab.guguchia@psi.ch; esantos@ed.ac.uk

Competing electronic phases underlie a number of unusual physical phenomena in condensed matter[1–3]. From superconductivity up to ferromagnetism, when the competition is sizeable a common outcome is phase separation. Compounds that have shown such behaviour are mostly of complex magnetic structures including cuprates[1], iron-based superconductors[3], ruthenates[2], topological kagome magnets[4], and manganites[5,6]. A contrasting case is found in the layered transition metals[7–11] where the presence of heavy halide atoms, like in CrI$_3$, stabilises pronounced anisotropy constants resulting in what appears a homogeneous ferromagnetic phase without any separation[12,13]. Nevertheless, recent experiments[14–17] have unveiled the presence of many subtleties in the magnetism of this compound which a single magnetic transition and a structural phase fail to capture.

Firstly, the magnetic properties of CrI$_3$ depend sensitively on the system structure. It is now understood that whereas bulk CrI$_3$ is ferromagnetic (FM) at 61 K[18] with presumably a rhombohedral stacking[14,15,19], thin layers can exhibit antiferromagnetic (AFM) coupling at 45 K in monoclinic[14,15]. However, the monoclinic phase is only observed in bulk at high temperatures. The origin of this puzzling behaviour has not been reconciled since the birth of the field of 2D vdW magnets[20,21]. Secondly, multiple anomalies can be observed in the temperature dependence of the magnetic susceptibility of bulk CrI$_3$ below 61 K[16,17,19]. Such anomalies imply that a more complex magnetic ordering involving spins not directly aligned with the easy-axis is likely emerging. Moreover, recent Raman measurements[22] appear contradictory with the appearance of a rhombohedral phase for thin layers in both FM and AFM ordering. This is followed by an anomalous phonon mechanism with the deviation of the linewidths as the temperature decreases[22]. Whether different magnetic phases may exist or competition occurs between structural phases is largely unknown. Here we systematically study the evolution of the magnetic and crystal structures of CrI$_3$ under different temperatures through a synergy of compelling techniques. Such approach has resulted being instrumental to identify, characterize and understand the distinct macroscopic ground states observed in this vdW material with competing magnetic and structural phases.

## Results

**Microscopic details of different magnetic phases: $\mu$SR experiments.** In a $\mu$SR experiment, positive muons implanted into a sample serve as extremely sensitive local microscopic probes to detect small internal magnetic fields and ordered magnetic volume fractions in the bulk of magnetic systems. See details on Supplementary Notes 1-2. Zero-field $\mu$SR time-spectra are recorded in a powder sample of CrI$_3$ below (5 K, 30 K, 54 K, and 60 K) and above (65 K and 80 K) the magnetic ordering temperature (Fig. 1a, b). A paramagnetic state is generally characterised by a small Gaussian Kubo-Toyabe depolarization of the muon spin originating from the interaction with randomly oriented nuclear magnetic moments. Conversely, the spectra from 150 K down to 62 K, exhibit a relatively high transverse depolarization rate $\lambda_T \simeq 4.9(2)\,\mu s^{-1}$. This reflects the occurrence of dense electronic Cr moments and indicates strong interactions between them. In this scenario a novel correlated paramagnetic state may be present in the system at temperatures above the actual Curie temperature.

As the crystal is cooled down, in addition to the paramagnetic signal, an oscillating component with a single well-defined frequency is observed at $T \lesssim 61$ K (Fig. 1a, b). Below 50 K, a spontaneous muon spin precession with two well-separated distinct precession frequencies is observed in the $\mu$SR spectra and persists down to 5 K. The temperature dependences of the

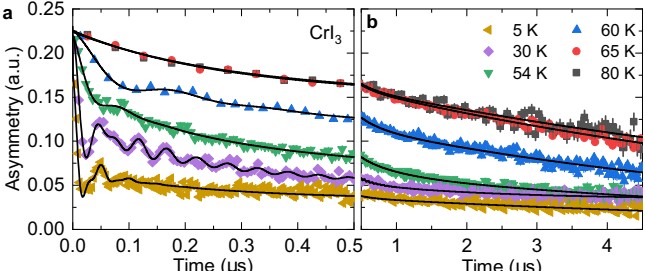

**Fig. 1 $\mu$SR spectroscopy applied to CrI$_3$. a, b** Zero-field $\mu$SR spectra recorded at various temperatures for the polycrystalline sample of CrI$_3$ shown in the low and extended time interval. The solid lines are the fit of the data using the methods of Supplementary Note 2. Error bars are the standard error of the mean in about ~$10^6$ events. The error of each bin count is given by the standard deviation of $n$. The errors of each bin in the $\mu$SR asymmetry are then calculated by statistical error propagation.

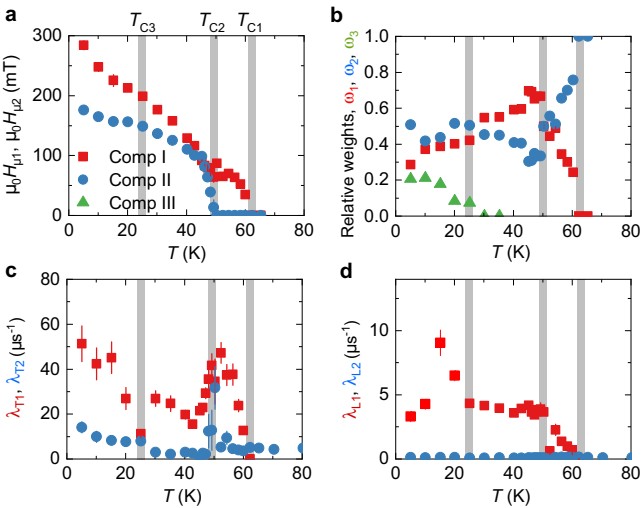

**Fig. 2 Temperature dependent $\mu$SR parameters. a** The temperature dependence of the internal magnetic fields for the observed two components in CrI$_3$. **b** The temperature dependence of the relative weights $\omega_{1,2,3}$ of the three components in the total signal. **c, d** The temperature dependence of transverse depolarization rates $\lambda_{T1}$, $\lambda_{T2}$ and the longitudinal depolarization rates $\lambda_{L1}$, $\lambda_{L2}$ for two components, respectively. The error bars represent the standard deviation of the fit parameters.

internal fields ($\mu_0 H_\mu = \omega / \gamma_\mu^{-1}$) for the two components are shown in Fig. 2a. The low frequency component shows a monotonic decrease and disappears at $T_{C2} = 50$ K. The high frequency component decreases down to 50 K, above which it keeps a constant value within a few Kelvin's range and then decreases again to disappear at $T_{C1} = 61$ K. Thus, the two oscillatory components have clearly different transition temperatures. This implies the presence of two distinct magnetic transitions in CrI$_3$. We also notice that an upturn on both $\mu_0 H_{\mu,1}$ and $\mu_0 H_{\mu,2}$ is seen below $T_{C3} = 25$ K. Moreover, a strongly damped component appears below $T_{C3}$ which is seen as some lost of initial asymmetry of the zero field $\mu$SR signal. This suggests the presence of another magnetic transition at this temperature. The temperature dependences of the relative weights of the individual components in the total $\mu$SR signal are shown in Fig. 2b. The weight of the high frequency component (component I) $\omega_1$ gradually increases below $T_{C1}$ and reaches maximum at $T_{C2}$, below which the second frequency appears. The third component raises below $T_{C3} = 25$ K. The components I and II share the weight of $30 - 70\%$ in the

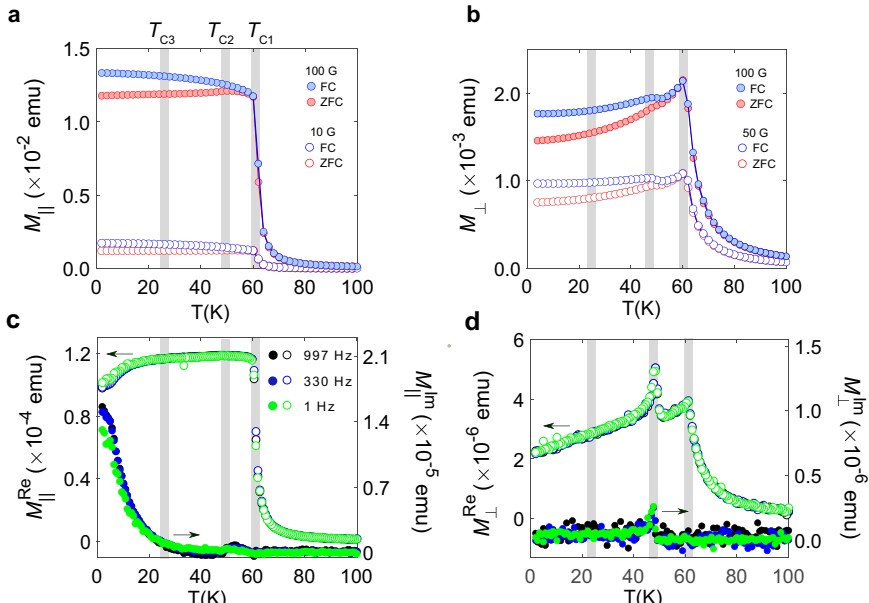

**Fig. 3 SQUID magnetometry. a–b**, Zero field cooled (ZFC) and field cooled (FC) temperature dependences of the DC magnetization $M$ at external magnetic fields (10 G, 50 G, 100 G) aligned parallel ($M_{\parallel}$) and perpendicular ($M_{\perp}$) to the crystallographic $c$-axis, respectively. The grey-shaded regions indicate the critical temperatures ($T_{C1}$, $T_{C2}$, $T_{C3}$) where the phase transitions occur as observed on the $\mu$SR measurements. The solid lines are a guide to the eye. **c–d**, Zero-field temperature dependences of the AC magnetization at parallel and perpendicular orientations, respectively. Three different frequencies (997 Hz, 330 Hz, 1 Hz) are used for plotting the real ($M_{\parallel}^{Re}$, $M_{\perp}^{Re}$) and imaginary ($M_{\parallel}^{Im}$, $M_{\perp}^{Im}$) parts of the AC magnetization.

temperature range between 30 K and 50 K. These results portray a clear coexistence of magnetically ordered phases in the temperature domain.

Figure 2 c, d shows the temperature dependences of the transverse $\lambda_T$ and the longitudinal $\lambda_L$ depolarisation rates, respectively, of components I and II. The $\lambda_T$ is a measure of the width of the static magnetic field distribution at the muon site, and also reflects dynamical effects (spin fluctuations). The $\lambda_L$ is determined by dynamic magnetic fluctuations only. For both components, $\lambda_T$ is higher than $\lambda_L$ in the whole temperature range, indicating that magnetism is mostly static in origin. However, $\lambda_{L1}$ has a higher overall value than $\lambda_{L2}$, implying that the magnetic order with $T_{C1} = 61$ K contains more dynamics. The presence of three transitions are clearly substantiated by the anomalies, seen in $\lambda_T$ and $\lambda_L$ (Fig. 2c, d). Namely, the $\lambda_{T1}$ starts to increase below $T_{C1}$ and peaks at $T_{C2}$, then decreases and tends to saturate. Nevertheless, it increments again below $T_{C3}$. $\lambda_{T2}$ also exhibits an increase below $T_{C3}$. Similarly, $\lambda_{L1}$ goes to high values for $T < T_{C1}$, saturates at $T < T_{C2}$ and then enlarges again for $T < T_{C3}$, followed by a peak at lower temperature.

We note that it is not possible to discriminate in the analysis the contribution of strongly damped components and a high frequency component into $\lambda_{L1}$ for $T < 30$ K and thus its peak at low temperatures could be due to the contribution from feature III. The increase of the dynamic longitudinal muon spin depolarization rate for $T < 30$ K, accompanied by a peak at lower temperatures, is a signature of a slowing down of magnetic fluctuations. These results imply that magnetic transitions at $T_{C1}$, $T_{C2}$ and $T_{C3}$ are influencing each other and they are strongly coupled, in spite of the fact that they are phase separated. Even though temperature is the main driving force to the appearance of these three phase transitions, their origin as well as their influence on the underlying magnetic properties of CrI$_3$ are still open questions to be further investigated. These findings point to a unconventional thermal evolution of the magnetic states in a 2D vdW magnet.

**Macroscopic magnetic properties: SQUID magnetometry**. To support the picture of multiple magnetic phases in CrI$_3$, we carried out SQUID magnetometry measurements on poly-crystalline and single crystal samples (Fig. 3). See Supplementary Note 3 for details. The magnetisation was measured at zero-field-(ZFC) and field-cooled (FC) conditions where the sample was cooled down to the base temperature in a weak external field and magnetization recorded upon warming. The most prominent anomaly in the thermal variation of the magnetic susceptibility onsets at 61 K as shown by DC measurements reflected in Fig. 3a, b. However, we also find signatures of additional magnetic transitions with distinct characteristics under different orientations of the magnetic field. Remarkably, there is a shoulder in both the FC and ZFC traces at around 50 K, which shows up very prominently in the in-plane magnetisation of the crystal (Fig. 3b) and much more subtly in the out-of-plane orientation (Fig. 3a).

The real part of the AC measurements (Fig. 3c, d) shows that the 61 K transition is independent of the AC drive frequency. Interestingly, there is no peak in the imaginary component of the AC magnetization in both orientations. This transition has so far been treated as a ferromagnetic long-range order phase transition[19]. However, the insensitivity of the imaginary component questions the type of magnetic order on the system. The second feature at 50 K is particularly visible in the thermal dependence of the in-plane AC magnetic moment of the crystal (Fig. 3d). This can be attributed to the second magnetic phase transition at $T_{C2}$. The relative height of this feature with respect to the main 61 K transition (Fig. 3d) is maximum at lowest fields. As with the main anomaly, this peak also exhibits no frequency dependence, which indicates that this phase transition is of long-range order nature. The DC magnetisation for the in-plane orientation at low temperature is non-zero, implying that some component of the magnetization exists in the crystallographic $ab$-plane. For both field orientations, the hysteresis is nearly zero from 61 K down to 50 K, whereas it suddenly increases below 50 K. This observation remarks the notion of a strong magnetic

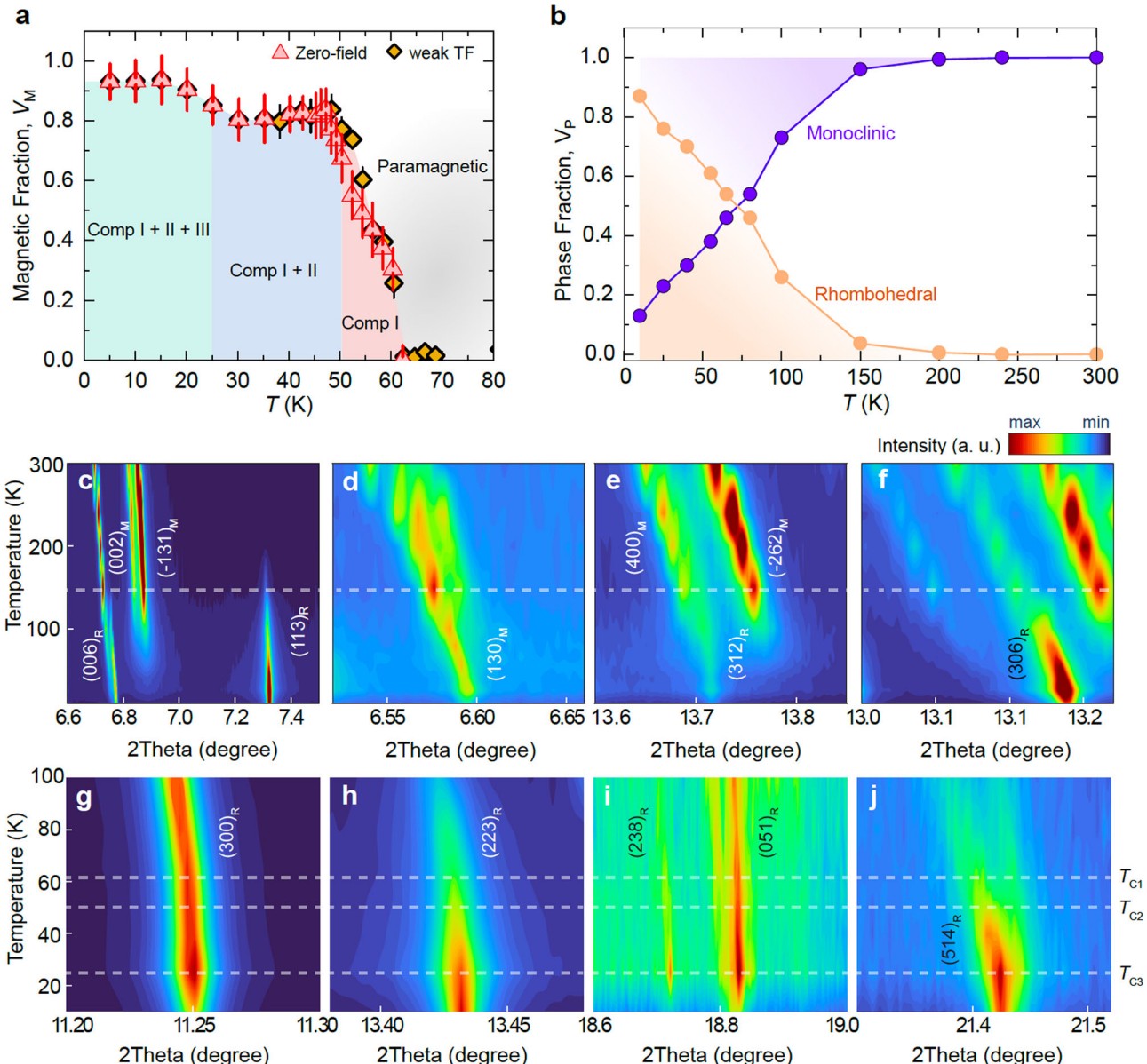

**Fig. 4 Phase diagram of the various magnetic and structural phases in CrI₃. a**, The temperature dependence of the total magnetic volume fraction $V_M$ determined at accurate weak transverse field (weak TF) and zero field $\mu$SR measurements. In the weak TF experiment, a small magnetic field of 30 G is applied nearly perpendicular to the muon spin polarisation. The different components seen in Fig. 2 are highlighted in each region of the temperature range with the paramagnetic phase above the Curie temperature. The error bars represent the standard deviation of the fit parameters. **b** The temperature dependence of the total phase fraction $V_P$ involving monoclinic (M) and rhombohedral (R) structures obtained via Synchrotron X-ray powder diffraction (SXRPD) measurements. **c–f** Temperature evolution below 300 K around selected monoclinic reflections $(002)_M$, $(-131)_M$, $(130)_M$, $(400)_M$, $(-262)_M$. Indexation of the peaks referred to the C2/m cell. Rhombohedral reflections $((006)_R$, $(113)_R$ $(312)_R$, $(306)_R)$ are also included for comparison showing the coexistence of both monoclinic and rhombohedral phases throughout the entire temperature range. The dashed line at ~150 K highlights the increase (decrease) of rhombohedral (monoclinic) phase. **g–j** Temperature evolution below 100 K of the SXRPD contour plot around the rhombohedral reflections $(300)_R$, $(223)_R$, $(238)_R$, $(051)_R$, $(514)_R$ in CrI₃. Indexation of the peaks referred to the R$\overline{3}$ cell. Dashed lines mark the successive magnetic transition temperatures ($T_{C1}$, $T_{C2}$, $T_{C3}$) observed through $\mu$ − SR in **a**.

anisotropy along the *c*-axis for CrI₃, and reveals the presence of some in-plane magnetic moment.

Moreover, the imaginary component of the AC magnetisation for the in-plane orientation in Fig. 3d exhibits a slight increment below ~ 30 K with a reduction of the real component. The fact that the most significant effect across ~ 30 K was seen in the imaginary part of the AC susceptibility indicates that the transition at this temperature is related to the slow in-plane magnetic fluctuations. These results lay the foundation of three different temperature phase domains, which are consistent with the $\mu$SR results and can

be considered as an independent piece of evidence for the presence of multi magnetic phases in CrI₃.

**Coexistence of structural phases: Temperature-dependent synchrotron X-ray diffraction.** The behaviour observed on the critical temperatures involves a volume-wise interplay between various magnetic states, providing an important constraint on theoretical models. One possible interpretation of the data is that below $T_{C1}$ there is an evolution of the magnetic order in specific volumes of the crystal, which coexists with a correlated

paramagnetic state. This interpretation is supported by the temperature dependent measurements of the total magnetic fraction $V_m$ (Fig. 4a). The magnetic fraction $V_m$ does not acquire the full volume below $T_{C1} = 61$ K. Instead, it gradually increases below $T_{C1}$ and reaches ~ 80% at $T_{C2} = 50$ K. An additional increase of $V_m$ by $10 - 15\%$ takes place below $T_{C3} = 25$ K, at which the third strongly damped component appears and reaches nearly 100%. The magnetism below $T_{C3}$ does not give extra coherent precession but it causes the strong depolarization of the $\mu$SR signal, reflected in the lost of the initial asymmetry. This indicates that $10 - 15\%$ volume is characterised by highly disordered magnetic state.

The volume-wise evolution (Fig. 4a) of the magnetic order across $T_{C1}$, $T_{C2}$, and $T_{C3}$ in CrI$_3$ strongly suggests the presence of distinct magnetic states in separate volumes of the system. We quantify this via the volume fraction $V_P$ of the sample obtained from Rietveld refinement of the synchrotron X-Ray powder diffraction data (Fig. 4b). See details in Supplementary Note 4. We observe that the material is composed of a mixture of rhombohedral (R) and monoclinic (M) phases[19] on a broad thermal range. The quantification of $V_P$ confirmed the absence of a single-phase structural scenario below the first-order crystallographic transition temperature happening in our system at around 150 K. The high-temperature monoclinic phase (C2/m) is not entirely substituted by the low-temperature rhombohedral structure (R$\overline{3}$). Indeed, the two-phase coexistence region is not restricted to the narrow interval previously proposed[19]. We found evidence of the persistence of a residual volume of the sample in the monoclinic phase in all measured temperature range, down to our base temperature (10 K) (Fig. 4c-f). Most interestingly, some peak widths and intensities show significant discrepancies with that expected from the original structural dichotomic model[19]. For instance, the intensity of the monoclinic peak (130)$_M$ (Fig. 4d) is barely affected by the temperature over the whole spectrum. Conversely, other peaks such as (-131)$_M$ and (002)$_M$ (Fig. 4c), and (400)$_M$ and (-262)$_M$ (Fig. 4e), are gradually suppressed at lower temperatures but do not disappear completely. It is worthwhile highlighting their Lorentzian-shape with anomalously wide half-width indicating that the monoclinic phase persists down to low temperatures in the form of short-ranged domains not much bigger than hundreds of Angstroms in size. We observed that the two coexisting structural phases share essentially the same value for the $a$-parameter, i.e., $a = 6.844$ Å at 80 K in both monoclinic and rhombohedral, suggesting intergrowth and a composite-like microstructure. Still, the goodness of fit parameters of the Rietveld refinement at low temperatures (below 150 K) portrays significant discrepancies with the original structural models, precluding the extraction of more quantitative conclusions regarding the high-to-low temperature phase conversion and urging for a more in-depth revision of the crystal structure resolution. The origin of the continuous variation of the fractions of monoclinic and rhombohedral (Fig. 4a) over a broad range of temperature is rather unclear and calls for additional works on synthesis, characterisation and crystal phase modelling.

Interestingly, the most compelling temperature-dependent structural evolution concerns the rhombohedral reflections which describe characteristic temperature-dependent features that are closely correlated with the magnetic critical temperatures observed (Fig. 4g–j). Firstly, the evolution of some structural reflections suggests a sizeable increase of intensity near ~ 50 K, such as in (223)$_R$ (Fig. 4h). Secondly, a significant number of rhombohedral peaks (though not all) display a maximum of intensity at ~ 25 K and then an intensity drop is clearly observed below that temperature (Fig. 4g, i, j). These relative intensity changes depict significant structural changes in the crystal, and correlate well with the magnetostructural effects in CrI$_3$ around

the magnetic transitions $T_{C1}$, $T_{C2}$ and $T_{C3}$. Additional discussions are included in Supplementary Note 5 on several other reflection peaks at different temperatures which further support the coupling between structural phases and magnetism. It is worth mentioning that we have discarded the presence of concomitant phases by chemical analysis and by the instability of the refinement process in the presence of chromium oxides, hydrates and diiodide phases.

## Discussion

We would like to emphasise that while $T_{C1}$ has been previously observed[19], with some unclear evidences for $T_{C2}$[17], $T_{C3}$ has never been noticed with macroscopic probes. As mentioned above, we relate the transition across $T_{C3}$ to the slowing down of the spin fluctuations. The magnetic moments fluctuate at a rate which is slower than the nearly instantaneous time scale of other techniques, e.g., neutron scattering[23]. Thus, neutron scattering and specific heat would hardly be sensitive to $T_{C3}$. $\mu$SR as a local probe and sensitive to small ordered fraction and slow fluctuations, combined with AC susceptibility, uncovers the novel $T_{C3}$ transition. Furthermore, the full width at half maximum (FWHM) of the (1,1,0) peak in neutron diffraction on bulk CrI$_3$[23], shows a smooth increase below 60 K with sudden variations at 50 K and within the range of ~ $32 - 25$ K. This is similar to the temperature dependence of the magnetic fraction $V_M$ (Fig. 4a) extracted from $\mu$SR measurements. As a volume-integrating probe in reciprocal space, neutron scattering techniques are sensitive to both the ordered moment and its volume fraction, but these two contributions cannot be separated from the measured scattered intensity. Hence, some additional transitions bellow $T_{C2}$ can be missed in the temperature dependence of the neutron scattered intensity. In particular, when two magnetic states are in microscopic proximity with each other (phase separation). In $\mu$SR however we can separately measure the internal field and the ordered volume fraction. The local probe features of $\mu$SR make this technique an excellent complementary approach to neutron diffraction and magnetization measurements.

The strong interplay involving distinct structural phases and competing magnetic orders found in CrI$_3$ raised several implications on the understanding of past and ongoing investigations on this material. In light of our findings, it is not surprising that thin layer CrI$_3$ assumed a monoclinic structure and consequently an AFM ordering[12,14,15] as that is one of the phases stabilised in bulk. Indeed, when bulk CrI$_3$ is exfoliated in a glove-box at room-temperature, monoclinic is the phase present in the structure. This phase does not change to rhombohedral as several groups had observed in different samples, devices, and conditions[15,17,20,24–26]. The coexistence of rhombohedral and monoclinic in bulk CrI$_3$ may suggest that both FM and AFM couplings are present over the entire crystal with no preference whether layers are more exposed to the surface or internal to the system[24,27]. The mixing of both structural phases is also a strong asset for breaking the inversion symmetry in centrosymmetric materials[28] and consequently the appearance of chiral interactions (i.e., Dzyaloshinskii-Moriya)[23,29,30]. In systems where the competition between single-ion anisotropy and dipolar-filed is not substantial, geometrical faults may contribute to the appearance of topologically non-trivial spin textures. Furthermore, the observation of the coexistence of two structures in bulk CrI$_3$ provides an interesting framework for further theoretical and experimental investigations. Such as in terms of crystal prediction at different temperatures and phase-mixing (e.g., random structure search), stacking order organisation at low energy cost, and spin-lattice mechanisms for unknown magnetic phases.

## Methods

**CrI₃ bulk crystal growth**. Chromium triiodide crystals were grown using the chemical vapour transport technique. Chromium powder (99.5% Sigma-Aldrich) and anhydrous iodine beads (99.999% Sigma-Aldrich) were mixed in a 1:3 ratio in an argon atmosphere inside a glovebox. 972 mg of the mixture were loaded into a silica ampoule with a length, inner diameter and outer diameter of 500 mm, 15 mm and 16 mm respectively. Additional details in Supplementary Notes 1.

**μSR experiment and analysis**. The μSR method is based on the observation of the time evolution of the spin polarization $\vec{P}$(t) of the muon ensemble. In μSR experiments an intense beam ($p_\mu = 29$ MeV/c) of 100 % spin-polarized muons is stopped in the sample. Currently available instruments allow essentially a background free μSR measurement at ambient conditions[31]. Additional details in Supplementary Notes 2.

**SQUID magnetometry**. Magnetization curves and zero-field-cooled/field-cooled susceptibility sweeps were carried out in a SQUID magnetomoter (Quantum Design MPMS-XL-7) on single crystals of CrI₃ were the relative orientation of the basal plane of the sample with the external magnetic field (both AC and DC) is controlled. Additional details in Supplementary Notes 3.

**Synchrotron X-ray diffraction measurements**. Synchrotron X-ray powder diffraction (SXRPD) measurements were performed at the the new Material Science Powder Diffraction beamline at ALBA Synchrotron. Powder Diffr., 28, S360–S370 (2013), proposal no. 2021014858. (Barcelona, Spain) using the multi-crystal analyser MAD detector system. Additional details in Supplementary Notes 4.

## Data availability

The data that support the findings of this study are available within the paper and its Supplementary Information.

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

## Acknowledgements

μSR experiments were performed at at the πM3 beam line (low background GPS instrument) of the Swiss Muon Source (SmuS) of the Paul Scherrer Insitute, Villigen, Switzerland, under proposal ID: 20190297 with EJGS as the PI. GT thank Prof. Alexander Shengelaya and the Georgian National Science Foundation (grant PHDF-19-060) for funding support to participate in μSR experiments led by ZG and EJGS. EJGS acknowledges computational resources through the CIRRUS Tier-2 HPC Service (ec131 Cirrus Project) at EPCC funded by the University of Edinburgh and EPSRC (EP/P020267/1); ARCHER UK National Supercomputing Service (http://www.archer.ac.uk) via Project d429. JLGM acknowledges the Spanish Ministerio de Ciencia, Innovación y Universidades for funding support through Project RTI2018-098537-B-C21, cofunded by EU ERDF program, and the "Severo Ochoa" Programme for Centres of Excellence in R&D (grant CEX2019-000917-S, FUN-FUTURE). EJGS acknowledges the Spanish Ministry of Science's grant program "Europa-Excelencia" under grant number EUR2020-112238, the EPSRC Early Career Fellowship (EP/T021578/1), and the University of Edinburgh for funding support. ENM acknowledges the European Research Council (ERC) under the Horizon 2020 research and innovation programme (ERC StG, grant agreement No. 803092) and to the Spanish Ministerio de Ciencia, Innovación y Universidades for financial support from the Ramon y Cajal program (Grant No. RYC2018-024736-I). This work was also supported by the Spanish Unidad de Excelencia "María de Maeztu" (CEX2019-000919-M). CP is thankful for the financial support of the Spanish Mineco Project No. FIS2017-83295-P.

## Author contributions

E.J.G.S. conceived the idea and supervised the project. Z.G., J.M.S., H.L., and G.T. performed the μSR experiments. E.N.M., J.M.S. undertook the SQUID characterisation and prepared the samples. C.P. performed the X-ray measurements, and helped in the analysis together with E.N.M., J.M.S., J.L.G.M. E.J.G.S. helped on the analysis, prepared the figures and wrote the paper with inputs from all co-authors. All authors contributed to this work, read the manuscript, discussed the results, and agreed to the contents of the manuscript.

## Competing interests

The authors declare no competing interests.
