## [Peer Review File · Nature Communications]

REVIEWER COMMENTS

Reviewer #1 (Remarks to the Author):

The authors report μ SR and magnetization measurements, and density functional theory calculation on CrI₃, a ferromagnetic van der Waals material that can be cleaved into a monolayer and have many interesting properties. There are many studies of this compound before, the key new discovery of the work is the claim that ferromagnetic phase transition at 61 K in this compound only involves 25% volume fraction of the material that is ferromagnetically ordered. Upon cooling further to lower temperatures, the system exhibits two additional phase transitions at 50 K and 25 K that drive the system to 80% and nearly 100% magnetic ordered volume fraction. The authors used magnetic susceptibility data and density functional theory calculation to back up the μ SR results, and claim that magnetic order in this system is quite different than initially thought.

While these results are interesting and can potentially be published, I think there are a lot of additional work that the authors have to do in order to establish their case. Let me list the major problems I have with the paper below, and see how authors can address them.

1. It is well known that system only displays a tiny *c*-axis lattice distortion across T_c around 61 K (from Ref. 9). If the authors claim that only 25% of the sample is ordered below 61 K, they should identify the structure of the phase that exhibits magnetic order, and those that does not. Without doing this, how do one know which phase is important for magnetism. From X-ray scattering experiment, it seems there is only one rhombohedral phase below about 150 K. Are the authors claim that only a part of the rhombohedral phase is ordered magnetically? If so, why? In typical phase separation materials, the fundamental underlying structure is anisotropic or inhomogeneous. In the cited papers for different classes of materials, there are intrinsic lattice or electronic anisotropy. This does not seem to be the case for CrI₃, and therefore the proposed three magnetic phase transition scenario is surprising and needs to be justified by experimental facts.

2. The mixed magnetic/nonmagnetic phase scenario claimed by the authors near T_c is not new. In a recent neutron scattering experiment, Chen et al. (Phys. Rev. B 101, 134418 (2020)) showed that the ferromagnetic phase transition in CrI₃ is weakly first order phase transition, which is the same as claiming that the ferromagnetic order does not form uniform phase below T_c . Furthermore, the authors have shown that system has no critical scattering, and magnetic correlation length is not resolution limited at 10 K. However, from looking at the magnetic order parameter and spin-spin correlation data, I don't see additional phase transitions at 50 K and 25 K. If the authors are claiming that there are these additional phase transitions, they should present microscopic evidence to demonstrate the spin orientation of these additional phase transitions, instead of μ SR fits to oscillation which may or may not represent the real situation. Unfortunately, μ SR cannot give these information, and the lack of such information makes these claims difficult to believe.

3. If the authors are indeed correct about the additional phase transitions at 50 K and 25 K, how come there is no thermal dynamic evidence for them? From the heat capacity measurements of Ref. 9, there is only one peak around 61 K, and I don't see any anomaly across 50 K and 25 K. If the authors are correct about the phase transition at 50, which supposedly involves 40% volume fraction of the system, how can there be no signature in heat capacity anomaly?

4. The magnetic susceptibility measurements are interesting, but I am not sure how a second magnetic phase transition can be so obvious with AC susceptibility but not with DC susceptibility. This needs to be clearly explained. Do authors believe that this second magnetic phase transition arises from the parts of the sample that are still paramagnetic below 61 K? If so, is this a second order phase transition and what is the magnetic structure?

5. The third magnetic phase transition is also strange. If the authors really believe that the system have these three magnetic phase transitions, they should do measurements to sort out what are these phases. Are they all occur within the standard rhombohedral lattice structure or there are some subtle differences in crystal structure that resulted in these different phases. I know these are tough questions that the authors may or may not be able to address. But they should at least provide discussion on possible scenarios, otherwise the paper reads like mostly a speculation paper rather than a solid scientific accomplishment.

6. In the theoretical simulation, the authors proposed possible magnetic structures of the different phases, but there is no microscopic justification as to why this might happen. If the underlying lattice structure is identical, why the system prefers to have these three phase transitions when there is only one underlying lattice structure.

In summary, while the reported results are interesting, there are considerable problems with the paper as written. The authors should carry out new experiments to sort out issues raised above before the paper can be accepted for publication. Otherwise they should consider submit the paper elsewhere.

Reviewer #2 (Remarks to the Author):

The authors present experimental evidence for several magnetic phase transitions in bulk CrI₃, based on muon spin rotation and SQUID measurements. In addition to the "standard" ordering transition at 61 K two additional phase transitions at 50 K and 25 K are identified. The interpretation of the results are based on macroscopic spin dynamics.

I find the work highly interesting, but I think the theoretical interpretation of the different phases is insufficient. First of all, the methodology of the spin dynamics is unclear. What is meant by the number of spins pointing along some direction for example? I assume the analysis is classical and in that case the spins should be vectors? Moreover, the experimental data quite clearly shows that something interesting is going on at 50 K and 62 K (the transition at 25 K is not as obvious), but it is not at all clear from the theoretical treatment what is going on. The authors mention different degrees of disorder, but the descriptors for disorder seems quite arbitrary and I cannot see how these give rise to actual phase transitions as indicated by the experimental data (and title of the manuscript). One would always expect high degree of disorder when closing in on the ordering temperature (for example at 50 K and 62 K) so I do not see how it is surprising. Is it because the authors expect highly ordered domains, which themselves destroy long-range order at these temperatures?

I am also surprised that the authors have not addressed the issue of intralayer vs interlayer order. To me that would be the initial guess for the different transitions. The one at 50 K marks the onset of intralayer order whereas the one at 62 K marks the onset of interlayer order. But that does not seem to be the case?

In summary I find the reported observations highly interesting, but the lack of a convincing interpretation of the different phases is a major drawback. For this reason I cannot recommend publication in Nature communications in its present form.

Reviewer #3 (Remarks to the Author):

The manuscript, 'Thermal disorder driven magnetic phases in van der Waals magnet CrI₃', reports two novel magnetic phase transitions in bulk CrI₃ below the 61 K Curie temperature. With μ SR and SQUID measurement, the authors determine the critical temperatures of both transitions and estimate the percentages of magnetically ordered volume at each magnetic phase. These results indicate that bulk CrI₃ has more complicated magnetic structures compared to its few-layer counterparts, where volume-wise competing electronic phases and thermal disorder play important roles. Overall, the results are convincing and inspiring, and I would recommend the work to be published on Nature Communication after the authors give further explanation on the following points.

The μ SR measurement is performed on CrI₃ powders while the SQUID is performed on CrI₃ single crystal. However, the manuscript does not give a reason for using different types of samples in these two measurements. One concern to me is that in powder samples the CrI₃ easy axis may be oriented in different directions. If so, the perpendicular local magnetic field felt by the muon spin will have different strength even in the same magnetic phase, which may result in multiple precession frequencies. The authors should explain how such complexity is avoided.

The manuscript attributes the longitudinal λ_L depolarization rates to CrI₃ spin fluctuations. Spin fluctuations usually peak around the critical temperature. However, in Fig.2d such peaking behavior is not seen at all three critical temperatures. Instead, a dip of λ_L is observed at Tc₂ and a peak of λ_L is observed below Tc₃. These behaviors would be better understood if the authors elaborate the influence of CrI₃ spin fluctuation on the muon spin depolarization.

The determination of Tc₁ and Tc₂ is quite convincing as clear features are observed at these temperatures in both μ SR and SQUID measurement. However, the determination of Tc₃ is a little confusing. In SQUID measurement there is almost no feature at the labeled Tc₃ region. In μ SR measurement the only prominent feature at Tc₃ is the single data point in Fig.2c. Meanwhile, below 20K (~10K) some emerging features (peak/dip/kink) could be detected in Fig.2b-d and Fig.S2c. I think more detailed analysis is needed to fully understand the magnetic transition around 20K.

The manuscript evaluates the magnetic volume fraction from weak-TF μ SR measurements. I noticed that the magnetic volume is also involved in the model of ZF- μ SR process. Could the magnetic volume be extracted from the ZF- μ SR data? If so, the authors may plot the result together with the data in Fig.3 to see if the two analysis are consistent. It would also help understanding if the authors elaborate the difference between these two types of measurements.

Response to Reviewers Comments:

Reviewer #1 (Remarks to the Author):

The authors report μ SR and magnetization measurements, and density functional theory calculation on CrI₃, a ferromagnetic van der Waals material that can be cleaved into a monolayer and have many interesting properties. There are many studies of this compound before, the key new discovery of the work is the claim that ferromagnetic phase transition at 61 K in this compound only involves 25% volume fraction of the material that is ferromagnetically ordered. Upon cooling further to lower temperatures, the system exhibits two additional phase transitions at 50 K and 25 K that drive the system to 80% and nearly 100% magnetic ordered volume fraction. The authors used magnetic susceptibility data and density functional theory calculation to back up the μ SR results, and claim that magnetic order in this system is quite different than initially thought.

While these results are interesting and can potentially be published, I think there are a lot of additional work that the authors have to do in order to establish their case. Let me list the major problems I have with the paper below, and see how authors can address them.

Response 1: *We thank the Reviewer for the kind words regarding our manuscript. We have substantially expanded the results presented in our work to address all comments raised by the Reviewer and changed the manuscript accordingly. In particular, we have included new temperature-dependent synchrotron X-ray data that further supports our previous analysis based on μ -SR and SQUID measurements. A point-by-point revision is showed below, with the location referring to the revised manuscript (highlighted in red).*

1. It is well known that system only displays a tiny c-axis lattice distortion across T_c around 61 K (from Ref. 9). If the authors claim that only 25% of the sample is ordered below 61 K, they should identify the structure of the phase that exhibits magnetic order, and those that does not. Without doing this, how do one know which phase is important for magnetism. From X-ray scattering experiment, it seems there is only one rhombohedral phase below about 150 K. Are the authors claim that only a part of the rhombohedral phase is ordered magnetically? If so, why? In typical phase separation materials, the fundamental underlying structure is anisotropic or inhomogeneous. In the cited papers for different classes of materials, there are intrinsic lattice or electronic anisotropy. This does not seems to be the case for CrI₃, and therefore the proposed three magnetic phase transition scenario is surprising and needs to be justified by experimental facts.

Response 2: *We thank the Reviewer for the comments and suggestions. We have considered them seriously and undertake additional temperature-dependent synchrotron X-ray diffraction measurements to sort out the nature of the observed transitions. Experiments were performed at the ALBA Synchrotron Light Facility in Barcelona, Spain. Details are included in SI.*

As it can be seen in the new Figure 4, there is a coexistence of two structural phases (monoclinic, rhombohedral) below 150 K. The increment of rhombohedral is followed by the decrease on monoclinic which it is temperature-driven. Both phases cross each other just above $T_{c1}=61$ K with a volume fraction $V_p \sim 50\%$ at ~ 72 K. Once the temperature drops further at $T_{c2}=50$ K and $T_{c3}=25$ K, the rhombohedral phase becomes dominant although the

monoclinic phase is always present at sizeable magnitudes: $V_p \approx 38\%$ at T_{c2} , and $V_p \approx 24\%$ at T_{c3} . Indeed, the two-phase coexistence region is not restricted to the narrow temperature interval previously proposed in Ref. 12 but extends substantially over the entire magnetic phase. Hence these new results confirm the absence of a single-phase structural scenario in bulk CrI_3 . This follows the Reviewer's comments regarding anisotropic and inhomogeneous structures in phase separation materials.

Regarding the Reviewer's question, we observed that roughly $\sim 25\%$ of the system is magnetically ordered at T_{c1} . As discussed above, not only the rhombohedral phase is present but also the monoclinic. The former carries the ferromagnetic coupling, whereas the latter the anti-ferromagnetic one. Our data suggests that both orders may compete to each other decreasing the total magnetisation of the system. Our μ -SR dataset and SQUID measurements (Figure 3) corroborate this picture.

2. The mixed magnetic/nonmagnetic phase scenario claimed by the authors near T_c is not new. In a recent neutron scattering experiment, Chen et al. (Phys. Rev. B 101, 134418 (2020)) showed that the ferromagnetic phase transition in CrI_3 is weakly first order phase transition, which is the same as claiming that the ferromagnetic order does not form uniform phase below T_c . Furthermore, the authors have shown that system has no critical scattering, and magnetic correlation length is not resolution limited at 10 K. However, from looking at the magnetic order parameter and spin-spin correlation data, I don't see additional phase transitions at 50 K and 25 K. If the authors are claiming that there are these additional phase transitions, they should present microscopic evidence to demonstrate the spin orientation of these additional phase transitions, instead of μ SR fits to oscillation which may or may not represent the real situation. Unfortunately, μ SR cannot give these information, and the lack of such information makes these claims difficult to believe.

Response 3: We agree with the Reviewer that μ -SR alone cannot help us to determine the magnetic structure as a function of temperature. This was one of the main reasons to complement our study with SQUID experiments (DC, AC) and temperature-dependent synchrotron X-ray diffraction measurements, which the latter was used to resolve the structures. We would like also to emphasize that μ -SR serves as an extremely sensitive local probe for detecting microscopic details of the i) static magnetic order, ii) ordered magnetic volume fraction, and iii) magnetic fluctuations in CrI_3 which cannot be accessed by neutron scattering measurements. The local probe feature makes μ -SR an excellent complementary technique to neutron diffraction and magnetization measurements.

Regarding the absence of magnetic transitions at T_{c2} and T_{c3} in neutron scattering measurements. We relate the transition across T_{c3} to the slowing down of the spin fluctuations. Since neutron is a very fast technique (e.g. $\sim 10^{-12}$ – 10^{-14} s) relative to μ -SR ($\sim 10^{-8}$ – 10^{-6} s), it will hardly be sensitive to the transition across T_{c3} . μ -SR as a local probe and sensitive to slow fluctuations plus AC susceptibility (Figure 3c-d) can access the low- T magnetic temperatures. Moreover, there are clear structural modifications at T_{c2} and T_{c3} , which in combination with the slow speed of the μ -SR experiments and all details provided, depict a sound characterisation of those phase transitions.

We also noticed that while neutron scattered intensity (Figure 1e in Chen et al.) does not show anomaly across 50 K, the full width at half maximum (FWHM) of the (1, 1, 0) peak (Figure 4c in Chen et al.) shows an intriguing behaviour: a smooth increase below 60 K with sudden variations at 50 K, and within the range of ~ 32 – 25 K. This is similar to the temperature dependence of the magnetic fraction from μ -SR (Fig. 4a). As a volume-integrating probe in reciprocal space, neutron scattering techniques are sensitive to both the ordered moment and

its volume fraction, but these two contributions cannot be separated from the measured scattered intensity. Thus, some anomaly can be missed in the temperature dependence of the scattered intensity. In μ -SR however we can separately measure the internal field and the ordered volume fraction. Since two magnetic states are in microscopic proximity with each other (phase separation), neutron scattering as a volume-integrating probe can overlook some fine details of the magnetic state.

We have included a thorough discussion in the manuscript highlighting these features (pages 11-12), and also cited Chen et al. as mentioned by the Reviewer.

3. If the authors are indeed correct about the additional phase transitions at 50 K and 25 K, how come there is no thermal dynamic evidence for them? From the heat capacity measurements of Ref. 9, there is only one peak around 61 K, and I don't see any anomaly across 50 K and 25 K. If the authors are correct about the phase transition at 50, which supposedly involves 40% volume fraction of the system, how can there be no signature in heat capacity anomaly?

Response 4: Interpretation of our data is that below $T_{C1} = 61$ K there is an evolution of the magnetic order in specific volumes of the crystal, which coexists with correlated paramagnetic state. The second magnetic order thereby occurs within the magnetic regions below T_{C1} . This transition involves only 30 % of the volume and it is in microscopic proximity with another magnetic state. There is a close relationship between these additional magnetic transitions with the variation of the crystal structures between monoclinic and rhombohedral as observed in our high-resolution X-ray results. Indeed, we are very cautious with the interpretation of the dataset included in Ref. 12 (old Ref. 9). In particular on the diffraction model used to interpret their results. We were unable to fit our data using the simple cell model used Ref.12. There are several reflections observed in our X-ray data (2θ) associated to both the R-3 and C2/m crystal phases below 200 K (Figure 4). This might explain the absence of a peak in the specific heat across 50 K in Ref.12.

In addition, as noted in **Response 6** below and **Response 3** above, we relate the transition across T_{C3} to the slowing down of the slow spin fluctuations. Since neutron scattering is a very fast technique, it will hardly be sensitive to the transition across T_{C3} . Moreover, since it only involves 15 % of the volume, it will not be detected by specific heat. μ -SR as a local probe and sensitive with slow fluctuations plus AC susceptibility can access the low-T magnetic crossover.

4. The magnetic susceptibility measurements are interesting, but I am not sure how a second magnetic phase transition can be so obvious with AC susceptibility but not with DC susceptibility. This needs to be clearly explained. Do authors believe that this second magnetic phase transition arises from the parts of the sample that are still paramagnetic below 61 K? If so, is this a second order phase transition and what is the magnetic structure?

Response 5: It is true that the 50 K transition is most clearly seen in zero-field AC susceptibility measurements (Figure 3c-d). However, this transition is also clearly visible in DC susceptibility measurements as a small kink (Figure 3b). Moreover, for both field orientations (\parallel , \perp), the hysteresis is nearly zero from 61 K down to 50 K, while it suddenly increases below 50 K (figure 3b). No hysteresis for 50-61 K suggests anti-ferromagnetic (AFM) order in this temperature range. While large hysteresis below 50 K for both field orientations suggest the canted AFM structure with the net FM moment both along the c-axis and within the ab-plane. Some evidence of this 50 K transition was also presented in Ref. 23, even though not as clear as in our dataset.

One possible interpretation of our data is that below $T_{c1} = 61$ K there is an evolution of the magnetic order in specific volumes of the crystal, which coexists with correlated paramagnetic state. The second magnetic order thereby occurs within the magnetically ordered phase existing below T_{c1} with variations of the lattice structure as demonstrated in Figure 4c-j. In this case, contributions from the AFM order would come from the monoclinic phase.

5. The third magnetic phase transition is also strange. If the authors really believe that the system have these three magnetic phase transitions, they should do measurements to sort out what are these phases. Are they all occur within the standard rhombohedral lattice structure or there are some subtle differences in crystal structure that resulted in these different phases. I know these are tough questions that the authors may or may not be able to address. But they should at least provide discussion on possible scenarios, otherwise the paper reads like mostly a speculation paper rather than a solid scientific accomplishment.

Response 6: *As discussed above in Responses 2-4, bulk CrI_3 presents both rhombohedral and monoclinic phases at a broad range of temperatures. This phase coexistence explains several of the Reviewer's questions and comments that were not clear on the first version of the manuscript. These new results also create a pathway to explain ongoing problems in the literature. For instance, the presence of AFM coupling in thin layer CrI_3 , which has been an open problem up to date. We became aware that recent neutron scattering measurements (Ref. 33) backup our findings with the extraction of the first-nearest interlayer exchange coupling with AFM character as well as the presence of chiral properties (e.g. Dzyaloshinskii-Moryia interactions). The latter may also be explained by the presence of both rhombohedral and monoclinic phases which breaks the inversion symmetry of the lattice inducing topological features on the magnon dispersion.*

We have included additional comments on the manuscript regarding implications and interpretation of published results in the literature.

6. In the theoretical simulation, the authors proposed possible magnetic structures of the different phases, but there is no microscopic justification as to why this might happen. If the underlying lattice structure is identical, why the system prefers to have these three phase transitions when there is only one underlying lattice structure.

Response 7: *We thank the Reviewer for the comments and suggestions. The main microscopic justification for the atomistic simulations was initially based on the thermal disorder that may generate the three magnetic phase transitions in CrI_3 . For that, we assumed just one crystal structure, e.g. rhombohedral. Nevertheless, after we have carried out a fully detailed analysis via additional temperature dependent X-ray measurements, which resulted in the discovery of structural phase coexistence, we realised that the simulations were no more relevant for the overall story. Hence, we have decided removing the simulation results from the manuscript since they provide minor insights on the new findings. We hope the Reviewer would be fine with the removal.*

In summary, while the reported results are interesting, there are considerable problems with the paper as written. The authors should carry out new experiments to sort out issues raised above before the paper can be accepted for publication. Otherwise they should consider submit the paper elsewhere.

Response 8: *We thank very much the Reviewer for his/her enlightening comments, and suggestions for additional experiments in order to fully understand the new results in our manuscript. We are confident that the new finding of the absence of a single crystal phase on*

CrI₃ will attract interest from the community. This may provide several pathways to understand the magnetic properties of 2D magnetic materials in general.

Reviewer #2 (Remarks to the Author):

The authors present experimental evidence for several magnetic phase transitions in bulk CrI₃, based on muon spin rotation and SQUID measurements. In addition to the "standard" ordering transition at 61 K two additional phase transitions at 50 K and 25 K are identified. The interpretation of the results are based on macroscopic spin dynamics.

I find the work highly interesting, but I think the theoretical interpretation of the different phases is insufficient.

Response 1: *We thanks the Reviewer for the kind words regarding our manuscript, and for his/her comments. We have reviewed the manuscript accordingly to the Reviewer's considerations. A point-by-point revision is showed below, with the location referring to the revised manuscript.*

First of all, the methodology of the spin dynamics is unclear. What is meant by the number of spins pointing along some direction for example? I assume the analysis is classical and in that case the spins should be vectors? Moreover, the experimental data quite clearly shows that something interesting is going on at 50 K and 62 K (the transition at 25 K is not as obvious), but it is not at all clear from the theoretical treatment what is going on.

Response 2: *As mentioned in Response 7 to #Reviewer 1, after we have finely undertaken temperature-dependent synchrotron X-ray diffraction measurements, which revealed the coexistence of rhombohedral and monoclinic crystal phases on bulk CrI₃, we realised that the simulations were no more relevant for the overall story. Because our initial hypothesis that thermal disorder may cause the appearance of additional magnetic transitions was no longer valid. Hence, we have decided removing the simulation results from the manuscript since they provide minor insights on the new findings. We hope the Reviewer would understand this modification.*

Moreover, on the theoretical side, an interesting problem in terms of the mixing of both structural phases on CrI₃ has risen for future studies. Obviously, this is beyond the scope of the present manuscript since several developments in terms of: 1) crystal prediction at different temperatures (e.g. random structure search), 2) stacking order organisation at low energy cost, and 3) spin-lattice coupling mechanisms.

We have included additional discussions on the manuscript covering these subjects.

The authors mention different degrees of disorder, but the descriptors for disorder seems quite arbitrary and I cannot see how these give rise to actual phase transitions as indicated by the experimental data (and title of the manuscript). One would always expect high degree of disorder when closing in on the ordering temperature (for example at 50 K and 62 K) so I do not see how it is surprising. Is it because the authors expect highly ordered domains, which themselves destroy long range order at these temperatures?

Response 3: *We thank the Reviewer for the comments and question. In light of the new X-ray diffraction results, we can infer that the geometrical disorder induced by the absence of a single crystal phase on bulk CrI₃ would be one of the main ingredients for the observation of the new magnetic phase transitions. At such environment, magnetic domains would not be as*

homogeneous as in a single-phase material, and competition may occur with domains having an AFM order in monoclinic phase with those with FM order in rhombohedral.

I am also surprised that the authors have not addressed the issue of intralayer vs interlayer order. To me that would be the initial guess for the different transitions. The one at 50 K marks the onset of intralayer order whereas the one at 62 K marks the onset of interlayer order. But that does not seem to be the case?

Response 4: *As mentioned in Response 2 to #Reviewer 1, the monoclinic phase holds an interlayer AFM coupling, whereas rhombohedral generates a FM one. Both phases however assume an intralayer FM order. Our data suggests that the interlayer magnetic orders may compete to each other decreasing the total magnetisation of the system. Our μ -SR dataset and SQUID measurements (Figure 3) corroborate this picture.*

In summary I find the reported observations highly interesting, but the lack of a convincing interpretation of the different phases is a major drawback. For this reason I cannot recommend publication in Nature communications in its present form.

Response 5: *We thank the Reviewer for the thorough comments and suggestions. They motivate us to undertake additional investigations to unveil the main details of the three magnetic phase transitions observed in our data as well as their explanation. This has taken substantial amount of time, resources and analytical thinking. However, this big effort has resulted in a much more complete analysis of our findings with several new implications not considered initially. We believe that we have addressed all Reviewer's concerns and our manuscript can be finally accepted in Nature Communications.*

Reviewer #3 (Remarks to the Author):

1. The manuscript, 'Thermal disorder driven magnetic phases in van der Waals magnet CrI₃', reports two novel magnetic phase transitions in bulk CrI₃ below the 61 K Curie temperature. With μ SR and SQUID measurement, the authors determine the critical temperatures of both transitions and estimate the percentages of magnetically ordered volume at each magnetic phase. These results indicate that bulk CrI₃ has more complicated magnetic structures compared to its few-layer counterparts, where volume-wise competing electronic phases and thermal disorder play important roles. Overall, the results are convincing and inspiring, and I would recommend the work to be published on Nature Communication after the authors give further explanation on the following points.

Response1: *We thank very much the Reviewer for his/her positive feedback and for supporting the publication of our manuscript on Nature Communications. We have reviewed our work accordingly to the Reviewer's considerations. A point-by-point revision is showed below, with the location referring to the revised manuscript.*

2. The μ SR measurement is performed on CrI₃ powders while the SQUID is performed on CrI₃ single crystal. However, the manuscript does not give a reason for using different types of samples in these two measurements. One concern to me is that in powder samples the CrI₃ easy axis may be oriented in different directions. If so, the perpendicular local magnetic field felt by the muon spin will have different strength even in the same magnetic phase, which may result in multiple precession frequencies. The authors should explain how such complexity is avoided.

Response 2: The value of the muon spin precession frequency is determined only by the magnitude of the magnetic field vector. It does not depend on the angle between the internal field orientation and the muon spin polarisation. In this sense, the homogeneous magnetically ordered polycrystalline sample will give only one precession frequency. If the sample contains phase separated two distinct magnetic regions than two precession frequencies are expected. Since muons stop uniformly throughout a sample, the amplitudes of the μ -SR signals arising from the different regions of the sample are proportional to the volume of the sample occupied by a particular phase.

3. The manuscript attributes the longitudinal λ_L depolarization rates to CrI3 spin fluctuations. Spin fluctuations usually peak around the critical temperature. However, in Fig.2d such peaking behavior is not seen at all three critical temperatures. Instead, a dip of λ_L is observed at T_{c2} and a peak of λ_L is observed below T_{c3} . These behaviors would be better understood if the authors elaborate the influence of CrI3 spin fluctuation on the muon spin depolarization.

Response 3: We thank the Reviewer for his/her interesting point. The peak in λ_L is expected in two cases: (1) if the magnetic transition is of second order and there is quantum critical point related to it. (2) When there is a slowing down of magnetic fluctuations. In CrI₃, the smooth increase of the magnetic volume fraction below 61 K shows that the magnetic transition is of first order, in which case no peak of λ_L is observed across the transition. Now, regarding the peak in λ_L : As we discuss in the revised manuscript, the transition at $T_{c3} \approx 20$ K is related to the slowing down of magnetic fluctuations, which causes the increase of λ_L below $T_{c3} \approx 20$ K until the fluctuations freeze out and λ_L decreases at lower temperatures.

4. The determination of T_{c1} and T_{c2} is quite convincing as clear features are observed at these temperatures in both μ SR and SQUID measurement. However, the determination of T_{c3} is a little confusing. In SQUID measurement there is almost no feature at the labeled T_{c3} region. In μ SR measurement the only prominent feature at T_{c3} is the single data point in Fig.2c. Meanwhile, below 20K (~ 10 K) some emerging features (peak/dip/kink) could be detected in Fig.2b-d and Fig.S2c. I think more detailed analysis is needed to fully understand the magnetic transition around 20K.

Response 4: The transition T_{c3} is visible on both μ -SR and susceptibility measurements. From the magnetization data the transition around T_{c3} is only seen in zero-field AC susceptibility data for the orientation parallel to the c-axis, shown in Figure 4c. Namely, real part of the AC susceptibility shows the small reduction below $T_{c3} \approx 20$ K. However, imaginary part shows a large increase below T_{c3} . The fact that the most significant effect across T_{c3} was seen in the imaginary part of the AC susceptibility indicates that the transition at T_{c3} is related to the slow magnetic fluctuations. The increase of the dynamic longitudinal muon spin depolarization rate below T_{c3} , accompanied by a peak at lower temperatures, is also a signature of a slowing down of magnetic fluctuations. From μ -SR measurements, the transition at T_{c3} is also visible as increase of the total magnetic volume fraction by about 10-15 %. This new magnetic component does not give extra coherent precession, but it causes the strong depolarization of the μ -SR signal, reflected in the loss of the initial asymmetry. This indicates that 10-15 % volume is characterised by highly disordered magnetic state.

To summarise, we interpret the behaviour across T_{c3} in the following way: above T_{c3} sample contains 10-15 % of paramagnetic region with slow magnetic fluctuations, cooperating with the long-range magnetic components II and III. These fluctuations exhibit the slowing down below T_{c3} and a disordered static magnetic state is established at lower temperatures within this small volume fraction. Following the Reviewer's suggestion, the extended discussion about the transition at T_{c3} is included in the revised version of the manuscript.

5. The manuscript evaluates the magnetic volume fraction from weak-TF μ SR measurements. I noticed that the magnetic volume is also involved in the model of ZF- μ SR process. Could the magnetic volume be extracted from the ZF- μ SR data? If so, the authors may plot the result together with the data in Fig.3 to see if the two analysis are consistent. It would also help understanding if the authors elaborate the difference between these two types of measurements.

Response 5: *The relative fractions of the three components, extracted from the zero-field μ SR experiments are shown in Figure 2a. We also include the total magnetic fraction extracted from the zero-field μ -SR experiments along with the weak-TF data (Figure 4a). Data from these two experiments match fairly well with each other with almost no difference. This indicates that the results are not field-sensitive.*

REVIEWERS' COMMENTS

Reviewer #1 (Remarks to the Author):

I have read the revised manuscript and replies of the authors. I appreciate very much the efforts made by the authors in carrying out X-ray scattering experiments. The new results combined with muSR data are much more convincing, and I can now recommend the paper for publication in Nature Communications.

Reviewer #2 (Remarks to the Author):

The authors have done a significant amount of extra work to strengthen their case and now argues strongly that the novel phase transitions are the result of mixture of structural phases.

The nature of the phase transitions are, however, still rather unclear. From Fig. 4 it can be seen that the fractions of monoclinic and rhombohedral phases change rather continuously with temperature and I do not see any compelling explanation of the nature of the different phases. The authors simply argue that it is somehow related to structural domains.

The experimental evidence seems to be rather convincing though. And I suppose that it is not always possible to provide an immediate theoretical explanation of observations. I think the study could initiate several follow up studies that may clarify what is going on and due to the tremendous current interest in this material I believe the present study could be published in its present form.

I do think the authors should be more up front in telling the reader that the nature of the additional phase transitions is not clear. This is how I read the paper at least.

Reviewer #3 (Remarks to the Author):

I appreciate the authors' efforts for performing additional experiments and analysis in response to my comments. It's also enlightening to me reading the comments from the other reviewers. In addition to their previous version, the authors perform X-ray diffraction measurement on bulk CrI₃ by which they find the coexistence of the monoclinic and the rhombohedral phases below 61K. This observation motivates the authors to abandon their initial theoretical model and generate a microscopic picture consistent with the results from all three characterization methods. Extended analysis of the main features at the three magnetic transitions are also presented in the revised manuscript. The magnetic fraction volume derived from ZF- μ SR and TF- μ SR agrees surprisingly well, which makes their conclusion more convincing. In general, I am satisfied with the response from the authors as well as the changes made to the manuscript. I would like to recommend the revised manuscript to be published on Nature Communications.

Response to Reviewers Comments:

Reviewer #2 (Remarks to the Author):

The authors have done a significant amount of extra work to strengthen their case and now argues strongly that the novel phase transitions are the result of mixture of structural phases.

The nature of the phase transitions are, however, still rather unclear. From Fig. 4 it can be seen that the fractions of monoclinic and rhombohedral phases change rather continuously with temperature and I do not see any compelling explanation of the nature of the different phases. The authors simply argue that it is somehow related to structural domains.

The experimental evidence seems to be rather convincing though. And I suppose that it is not always possible to provide an immediate theoretical explanation of observations. I think the study could initiate several follow up studies that may clarify what is going on and due to the tremendous current interest in this material I believe the present study could be published in its present form.

I do think the authors should be more up front in telling the reader that the nature of the additional phase transitions is not clear. This is how I read the paper at least.

Response: *We thank the Reviewer for his/her kind words regarding our manuscript, and for accepting it in its present form.*

We have included new sentences earlier in the text (page 6, lines 103-105; pages 9-10, lines 176-179) pointing out the unknown nature of the additional phase transitions as suggested by the Reviewer.